# Functional Connectivity and Frequency Power Alterations during P300 Task as a Result of Amyotrophic Lateral Sclerosis

**DOI:** 10.3390/s21206801

**Published:** 2021-10-13

**Authors:** Claudia X. Perez-Ortiz, Jose L. Gordillo, Omar Mendoza-Montoya, Javier M. Antelis, Ricardo Caraza, Hector R. Martinez

**Affiliations:** 1Escuela de Ingeniería y Ciencias, Tecnologico de Monterrey, Monterrey 64849, Mexico; c.xochitl96@gmail.com (C.X.P.-O.); JLGordillo@Tec.mx (J.L.G.); mauricio.antelis@tec.mx (J.M.A.); 2Escuela de Medicina y Ciencias de la Salud, Tecnologico de Monterrey, Monterrey 64849, Mexico; rcaraza@tec.mx (R.C.); hector.ramon@tec.mx (H.R.M.)

**Keywords:** ALS, EEG, classifier, neural, connectivity, frequency-specific, BCI

## Abstract

Amyotrophic Lateral Sclerosis (ALS) is one of the most aggressive neurodegenerative diseases and is now recognized as a multisystem network disorder with impaired connectivity. Further research for the understanding of the nature of its cognitive affections is necessary to monitor and detect the disease, so this work provides insight into the neural alterations occurring in ALS patients during a cognitive task (P300 oddball paradigm) by measuring connectivity and the power and latency of the frequency-specific EEG activity of 12 ALS patients and 16 healthy subjects recorded during the use of a P300-based BCI to command a robotic arm. For ALS patients, in comparison to Controls, the results (*p* < 0.05) were: an increment in latency of the peak ERP in the Delta range (OZ) and Alpha range (PO7), and a decreased power in the Beta band among most electrodes; connectivity alterations among all bands, especially in the Alpha band between PO7 and the channels above the motor cortex. The evolution observed over months of an advanced-state patient backs up these findings. These results were used to compute connectivity- and power-based features to discriminate between ALS and Control groups using Support Vector Machine (SVM). Cross-validation achieved a 100% in specificity and 75% in sensitivity, with an overall 89% success.

## 1. Introduction

Amyotrophic Lateral Sclerosis (ALS) is one of the most aggressive neurodegenerative diseases causing the patient to lose the ability to move their muscles; it affects and kills upper and lower motor neurons [1]. Being a complex disease, the specific nature of the affectations is still unknown [1]. There is no particular detection method for ALS; thus, the detection procedure involves taking several tests to discard other diseases; this may be due to the lack of a biological marker or biomarker for ALS [1]. To understand ALS, more global holistic approaches have been undertaken. New research has indicated neurodegeneration in nonmotor areas too [2], and the disease has also been recognized as a multisystem network disorder characterized by impaired connectivity (a measure of how synchronized two brain regions are) [3,4,5,6,7]. Correlations have been found between changes in connectivity and cognitive scores from a neuropsychological battery (cognitive tests) [7]. Frontotemporal dementia [8] and cognitive disability [9] have been linked with ALS.

The use of different technologies gives some insight; for example, Positron Emission Tomography (PET) and Magnetic Resonance Imaging (MRI) studies have shown deterioration in motor cortex regions [3]. Alterations and differences in ALS are primarily found in functional Magnetic Resonance Imaging (fMRI) or similar costly procedures [3]. Correlations have been found, in studies with ALS patients, between Electro-encephalography (EEG) rhythms and MRI and transcranial magnetic stimulation (TMS) findings [10], and between fMRIs and EEG [11] in the past, suggesting that other neuroimaging findings could be replicated with EEG. EEG offers an understanding of the activity happening on the cerebral cortex originating from neural activity. It is also a portable, noninvasive brain imaging sensor that obtains cerebral information in real-time and generates responses [12]. This type of neuroimaging has been used as a detection method for neurodegenerative diseases. For example, analyzing the sharpness of the brain signals is commonly used to detect epilepsy [13]. Thus, EEG studies can show how the activity in an ALS brain is changing or degenerating, offering specialists further comprehension of a disease whose evolutive nature is still unknown. Even though some studies have been performed on ALS patients, they have usually focused on rest-state activity [5,7]. The potential use of these types of findings as a biomarker to detect ALS has even been assessed with a result of 100% in specificity but only 58% in sensitivity [4].

Testing during a cognitive activity offers additional information of a mental state. The P300 oddball paradigm tests cognitive activity through EEG. The oddball test is the presentation of repetitive stimuli randomly interrupted by a different stimulus. P300 is an Event-Related Potential (ERP) elucidated 300 ms after a stimulus. It is a signal that arises as a response of the brain to an external stimulus [14]. The visual P300 task involves communication between the parietal region and the frontal region [15], the frontal one being one of the most affected areas for ALS individuals [3]. P300 is commonly used in Brain–Computer Interfaces (BCIs) as its response is repetitive and detectable through the cognitive task. This quality permits P300 to act as a detection method. Additionally, the magnitude and latency of the P300 peak have shown a solid ability to detect other diseases such as Alzheimer’s, which is linked to a decreased peak and higher latency [16], and many other neurological disorders [14]. 

More recent studies have demonstrated the cognitive impairment of ALS individuals, especially with delayed latencies in P300 peaks [17,18,19]. These studies could offer a more robust understanding of the EEG if analyzed in the frequency domain as it offers a significant correlation to neural oscillations. Differences have been found in the functional connectivity and amplitude of Alpha and Beta frequency bands (9–13 and 14–30 Hz, respectively), suggesting a frequency-specific reduction in patterns in functional connectivity and amplitude [11,20], but a longitudinal analysis of the biomarkers found during the EEG studies among the same individuals has not been performed, in other words, showing that the advance of the disease has not been assessed until now. 

The present study provides insight into the neural alterations or changes occurring in ALS patients during a cognitive task (P300 oddball paradigm) by measuring EEG activity. The objective was to find the neural alterations, define these alterations as biomarkers, analyze their change in a longitudinal study in ALS patients, and classify between groups with them. Three numeric values were calculated (variables) from the time–frequency power signals to measure the EEG activity: the value of the peak power and its latency, and the connectivity between electrodes (a value between 0 and 1) for each frequency band. These variables were tested for a significant difference between ALS and Control groups; those who were found different were used to train a classifier that separates both groups and were further analyzed. 

## 2. Materials and Methods

Finding neural alterations in ALS patients with respect to healthy subjects is a method of obtaining insight into the mechanism that ALS follows. It could also serve as a detection aid to confirm or diagnose ALS. Different tests were made with the EEG data of 12 ALS patients and 16 healthy subjects to find these alterations. The data came from a 16 electrode-P300-based BCI system that was previously designed to aid ALS patients with communication with the outer world, such as moving a robotic hand orthosis [21]. 

The tests made to the data had the objective of obtaining a numeric value from the data, such as the time in milliseconds where the P300 peak was found or the power of that peak. The connectivity analysis calculated the connectivity (values from 0 to 1) between one electrode and the others. These three tests were made on the data decomposed in spectral power. A point-wise graphical analysis helped to analyze the 3D images from the time–frequency–power charts. Finally, the numeric values were taken as variables. The significantly different variables were further analyzed and some variables were selected as biomarkers. The biomarkers were used to train an SVM classifier or to observe an ALS patient’s evolution over time.

Finally, the numeric values were taken as variables. The significantly different variables were selected as biomarkers, and the biomarkers were used to train an SVM classifier.

### 2.1. P300-BCI System

The data used for this analysis were taken from the training stage of a P300-BCI designed for ALS patients. A P300-based BCI was previously developed with the purpose of assisting ALS patients to control a Hand of Hope robotic arm (Rehab-Robotics Company, China). Muscle movement loss is a common ALS symptom, hence deteriorating the original pathway for muscle movement; the objective of this BCI is to generate an alternative path for hand-muscles movement, as shown in Figure 1a. Instead of the brain moving the muscles of the hand, a WiFi-controlled hand orthosis forces the movement onto the hand [21].

For this test, subjects are comfortably seated in front of an LCD monitor in a silent room where only the testers and subject are present. An EEG cap is placed on the subject’s head, and gel is applied to the 16 selected electrodes (shown in grey in Figure 1a) and on the reference (which is attached to the right ear) by the testers. The Graphical User Interface (GUI) is composed of a picture of an open hand wearing the hand orthosis with one grey dot on top of each finger and another dot on the hand’s palm, and on the bottom, it has a rectangle for different instructions, depending on the stage. The task is based on the P300 oddball paradigm. The happy faces represent the oddball stimuli that will cause the P300 response or the ERP. This is the response the BCI is looking for to detect intentions. The first stage of the P300 experiments consists of training or calibration of the BCI system. With this training, the algorithm learns the subject’s P300 characteristics so that they can be identified later. 

The Training stage is composed of 8 blocks of training. Each block is composed of 5 stages, and the GUI’s state in each stage is shown in Figure 1b. First, in the Fixation stage, where the subjects must get ready for the experiment, a cross appears on the GUI’s rectangle for 2 s. Then, in the Target Presentation stage, one finger or the whole hand is indicated in the rectangle. The indicated finger shows which of the grey dots to observe in the Active Task stage, or if the whole hand appears indicated; then, the finger in the palm must be observed in the Active Task stage. Then, a second of Preparation is given, for the subject to prepare to begin the Active Task. Then, the Active Task stage begins. A happy face begins flashing in one dot at a time and the participant is asked to count in their head the number of happy faces that appear in the indicated spot while observing only the indicated dot. The face appears in a dot for 75 ms and then all dots are grey for 75 ms, repeating this pattern until the face appears in the indicated spot about 32 times. Finally, a 5 s rest is given to the subject, in preparation for the next block [21]. 

The Free Validation stage comes after the Training stage, where the subject is indicated to choose any desired dot, and as soon as the BCI detects the subject’s desired dot, it is colored red. This is repeated about 3 or 4 total times for the user to see that the BCI is following their instructions. Then, the Online Validation stage begins. This stage is very similar in procedure to the Training stage, except the BCI’s objective is to detect the dot the user is indicated to concentrate on. This block is repeated once per dot. Finally, the robotic arm is attached to the subject’s left hand by the testers and the Free Validation and Online Validations are repeated with it on. With the robotic arm attached, when the desired dot is detected by the BCI, the corresponding finger is contracted by the motors in the Hand of Hope. When the palm dot is detected, all the fingers are contracted. The results of these BCIs have already been reported in another paper [21]. 

The data from the Training stage were selected as it is the stage where more trials are available; with more trials, the signal-to-noise ratio is increased. To extract the trials, the moment where the happy faces or stimuli appear was extracted. The period from 300 ms before each stimulus until 700 ms after it was extracted for each trial. Only the trials where the happy face was located on the indicated (by the BCI) dot were taken into consideration for this analysis, as these trials were where the ERPs were present. A total of 264 trials were initially used for each patient.

### 2.2. Data Acquisition

The data were recorded by 16 monopolar electrodes positioned according to the 10–20 international system at positions in FZ, CZ, PZ, OZ, C1, C2, C3, C4, C5, C6, CP3, CP4, P3, P4, PO7, and PO8, as shown in Figure 1c, with the reference placed on the right earlobe and ground electrode at AFz. The electrodes were selected by the designers of the BCI with the objective of covering the motor cortex and the sites commonly used in P300 BCIs [21]. The signals were amplified using a g.USBamp amplifier (a g.GAMMASYS active wet electrode arrangement and a g.USBamp amplifier provided by g.tec medical engineering GmbH, Schiedlberg, Austria). The sampling rate was set at 256 Hz. The computer processed the EEG signals, displayed the GUI, synchronized and displayed stimuli, and sent control messages to the robotic arm. 

### 2.3. Subjects

The users of the experimental protocol were divided into two groups: ALS patients and the Control group. The ALS group contained 12 patients with Bulbar or Spinal ALS with mild to advanced levels of hand atrophy, six women and six men whose age had a mean of 59 ± 7. Additionally, 4 ALS patients went through the training more than once (with a minimum of three months between tests). These older training data were used to observe the evolution of selected variables. Only the oldest training data of each patient were used in the variable’s extractions. The ALS group was recruited from the patients attending the TecSalud ALS Multidisciplinary Clinic [21]. The Control group consisted of 16 healthy subjects, eight women and eight men whose ages were 33 ± 15.

### 2.4. Analysis

The trials obtained from the training stage were the basis of the analysis. The steps for the presented analysis are shown in Figure 2. These signals were preprocessed to obtain the most information out of them, and then two different studies were performed on each individual’s data: power and connectivity analysis. Latency and amplitude variables were extracted from the power analysis and the connectivity value between two electrodes was extracted from the connectivity analysis. All the variables obtained were compared between ALS and Control groups to select the best ones. Finally, the selected variables were observed over time in the ALS individuals and were used to train a classification model. 

### 2.5. Pre-Processing

A DC baseline correction was performed by averaging the activity from 200 ms before the P300 stimulus to time 0 (the specific time of the stimulus) and subtracting this value from every time point. An automatic trial rejection was performed based on three parameters: first, the maximum peak-to-peak value after the stimulus >200, as the brain signals being observed were between −100 and 100 mV [14]; then, the standard deviation of the trial after stimulus <50, and the noise to signal ratio >0.7. If any of these conditions is true, the trial is rejected. 

### 2.6. Power Analysis

For each trial, the EEG signal was extracted from 300 milliseconds (ms) pre-stimulus to 700 ms post-stimulus. It was decomposed via complex Morlet wavelet convolution with a set of wavelets ranging from 0.5 to 40 Hz and a variable number of cycles from 2 to 10. For each user and each electrode, the percentage change with a pre-stimulus base was calculated, so the power could be compared among frequencies and time. 

The power, *pow*, at each time point, *t*, is obtained by squaring the voltage, *v*, at each corresponding time point, as shown in Formula (1). The baseline interval selected was between 200 ms pre-stimulus and 0 ms (the moment of the stimulus). The value of the baseline, *R*, is the average power in this interval for each frequency band, *f*, as shown in Formula (2). Then, to obtain the activity, *A*, the voltage at each time point, *v*(*t*), is squared to obtain the power, *pow*(*t*), and an average is performed among all the trials, *tr* (for each subject), as shown in Formula (3). Finally, the power percentage change is obtained as shown in Formula (4). In this case, this was performed for each 0.75 Hz in the plot. Anything over 0 is considered a power increase or Event-Related Synchronization, and everything between below 0 is considered a power decrease or Event-Related Desynchronization [22]. The final powers were divided by neural bands (Delta [0.5–3 Hz], Theta [4–8 Hz], Alpha [9–13 Hz], and Beta [14–30 Hz]), and an average was performed among the frequencies of each band.
(1)$$pow\left(t\right)=v{\left(t\right)}^{2}$$
(2)$$R\left(f\right)=\frac{\sum_{i=ix}^{iy}pow\left(i,f\right)}{tp}$$
(3)$$A\left(f,t\right)=\frac{\sum_{x=1}^{tr}pow\left(i,f,\;t\right)}{tr}$$
(4)$$\mathrm{power}\;\%\;\mathrm{change}=\frac{\left(A-R\right)}{R}*100$$

### 2.7. Point-Wise Analysis

Complementary to the maximum values power analysis, a point-wise analysis was performed over the 3D graphs whose axes were time, frequency, and spectral power percentage change. Permutation-based statistics were performed to determine areas of significant difference. For this analysis, we assumed that there were no significant areas on the map. A null-hypothesis map was made by shuffling subjects among groups, taking the mean of each new group, and subtracting one map from another. Point by point (pixel by pixel), this operation was permuted 1000 times to create a distribution. Finally, the observed value was compared to the distribution obtained. The points whose observed value had a *p*-value below 0.05 were considered significant. 

### 2.8. Connectivity Analysis

Connectivity is a measure used to determine the oscillatory synchronization that exists between two brain regions, represented by electrodes. InterSite Phase Clustering (*ISPC*) is a connectivity measure that relies on the phase of the signals to determine the degree of connectivity. This theory is based on the concept that for two regions to be synchronized, they must be sending information and reading it at its maximum excitation point.

To calculate the *ISPC* values of each subject between signals from electrodes *x* and *y*, first, the signal is converted to an analytic signal; then, the analytic signal, as, is divided in frequency bands, *f*. The signal is then divided among the frequencies of each band, resulting in four signals, one for each band. Then, the signal’s instantaneous phases, *ph*(*t*,*f*), are calculated at each frequency band and each instant of time, *t*. An average of the differences (at each instant of time) between the two signals is obtained, as shown in Formula (5), and this is the *ISPC*. If the signals are synchronized, the value should be close to 1. The *ISPC* was calculated between every electrode and the other 15 electrodes.
(5)$$ISPC_{x,y}\left(f\right)=\bar{ph\left(t,f,x\right)-ph\left(t,f,y\right)}$$

Formula (5): How *ISPC* is obtained. 

### 2.9. Variables Extraction

Variables from the power and connectivity analysis were extracted. All variables mentioned were extracted for each subject and patient. From the power analysis, the values of the P300 peak and its latency were extracted. To compute these, the maximum value and its latency between 200 and 650 ms post-stimulus in the time–frequency data were calculated. From the connectivity, the *ISPC* value was used. In addition, 16 electrodes, 4 neural bands, and two conditions (magnitude and latency of P300 peak) resulted in 128 power variables; 16 electrodes compared with 15 electrodes in 4 neural bands resulted in 960 connectivity variables. All variables added to a total of 1088 total variables for each of the 28 subjects and patients. 

### 2.10. Separability Test

To find the variables that might be of interest among both groups (ALS and Control), a Wilcoxon rank-sum test (a nonparametric test that contrasts two samples in order to determine if they come from equally distributed populations) was performed for each variable. Only the variables significantly different were further observed (*p* < 0.05). 

Two approaches were made with the selected variables. First, we analyzed how the selected variables were represented in the original data. Second, we observed where patients’ older training data (the four of them that we have) stood among these selected data. 

### 2.11. Multiple Comparisons Correction

When multiple tests are being performed, a multiple-tests correction is needed. The reason is that in a normal distribution, we are expected to obtain some results that seem significantly different but occur because of chance. For the tests performed, two types of multiple-tests correction were performed. For the extracted variables analysis (maximum power, maximum power latency, and connectivity), a False Discovery Rate (FDR) was performed. For the graphical power analysis, an Extreme Point Correction was performed. 

#### 2.11.1. FDR Correction

The False Discovery Rate (FDR) is a method to fix the *p*-value when testing multiple comparisons, as in this case. The basis of this method is to compute the *p*-value of all nonsignificant results and use this value as a cutoff to make everything above it significant and below it nonsignificant, adjusting the *p*-value. While this is a graphical method, it can be performed through a mathematical approach. In this approach, first, all the *p*-values must be sorted from smallest to largest and ranked. The last value (top rank) is kept. The next largest value is the smaller between the previously adjusted *p*-value and the result of Formula (6) (where *p*(*r*) is the *p*-value of rank r of the current *p*-value, *n*_*p* is the number of *p*-values, and r is the rank *r*), and so on, until the smallest *p*-value.
(6)$$\mathrm{adjusted}\;p\text{-}{\mathrm{value}}_{r}=p\left(r\right)*\;\frac{n_{p}}{r}$$

Formula (6): Adjusted *p*-value

After performing this correction, the significance of all the discoveries was above 0.05, with the least value being 0.056, corresponding to the power results. 

#### 2.11.2. Extreme Point Correction

When performing graphical analysis, FDR is not the correct approach, as the number of points (or pixels) in the image affects the parameters of FDR, and intuitively this makes no sense, as a significant cumulus of points (or pixels) should remain significant independently of the resolution of the image. For this reason, an extreme point correction was performed. In this correction, the assumption is made that there are no significant areas on the map. A null-hypothesis map was made by shuffling subjects among groups, taking the mean of each new group, and subtracting one map from another. In this new graph, the least value and maximum values were extracted. The process was repeated 1000 times to create a bi-modal distribution. Then, everything above 0.025 and below 0.975 was considered not statistically important or different. Only the points outside of this area were still considered statistically different. 

## 3. Results

In this section, the results are presented and described. Additional figures were used to examine the distribution of groups variables. 

### 3.1. Power: Magnitude

The four bands were examined, each of which contained 16 electrodes. As a result, 64 variables were obtained for magnitude for each patient or Control. 

A significant difference was found only in the beta band. All of the ten variables that were found to have statistical differences were found in the same band. The electrodes that were found to be different between both groups are shown in Table 1. They were in the locations FZ, C2, CP3, and PO7 (*p* < 0.05), and C1, CZ, C4, P3, PZ, and P4 (*p* < 0.01). The locations of these specific electrodes are shown in Figure 3a.

#### 3.1.1. Distribution of Power Variables

In Figure 3b, the distribution of the power magnitude variables of each group, Control and ALS, can be seen. The distributions of electrodes found to be statistically different can be seen to have different interquartile ranges. The ALS median is below the Control median, showing a decrease in magnitude for the ALS group or a decreased activity. 

#### 3.1.2. ERPs

The ERP of the Beta band was computed to observe the differences remarked in the statistical analysis. As the ERPs show, in the Beta band, a peak can be seen at about 500 ms after the stimuli. The mean ERPs of channels located in the Central area (e.g., C1, CZ, C2, and C4) show opposite phases for both groups around 450 ms after the stimuli. This area is significantly different, as shown in Figure 4. In electrodes located on the Parietal area, the ALS group has lower peaks in P3, PZ, and P4. All of these variables (e.g., the peak value of spectral power in electrode CZ in the Beta band) are found amongst selected variables. For instance, in Figure 4, the ERP of channel CZ in the Beta band is shown. The Control average shows a positive peak around 400 ms, while the ALS average shows a negative deflection in this same time point. A negative peak where a positive peak should appear is usually due to an overactivation in another cerebral region. In contrast, in Figure 4, the PZ electrode, Beta band, both the Control and ALS group have a positive peak around 400 ms. However, the peak magnitude corresponding to the Control group is almost double that the ALS peak. This indicates a reduced activity in the beta band for the ALS group. 

These results can also be seen in the time–frequency charts. In Figure 5c,d, we can see channel PZ for the Control and ALS group. There is a clear difference in the magnitudes in all frequencies around 400 ms, especially around 20 Hz. The magnitude can be seen much higher for the Control group. In addition, in Figure 5a,b, the differences between groups are noticeable—the ALS group shows a spectral power decrement, where the Control group shows an increase. 

#### 3.1.3. FDR Correction

An FDR correction was needed, as mentioned before. All 1088 variables were no longer significant after FDR correction with *p* > 0.05. Power results had the lowest value after the FDR correction with a *p*-value of 0.056. To avoid false negatives (error type 2), further tests were performed.

#### 3.1.4. Point-Wise Analysis

After performing the point-wise analysis and the extreme pixel correction, the channels that were found to have an area of significant difference were C1, CZ, C2, C4, and OZ. In the central band, the effect seemed to dissipate as the channel became more distant from the central (Z or CZ) zone, which is on the center of the scalp in the central band, and underactivation was occurring for the ALS group. Another channel that had a significant difference, but was barely observed in previous results, was in channel OZ, almost at the same latency, as shown in the Figure 6.

### 3.2. Power: Latency

The four bands were examined, each of which contained 16 electrodes. As a result, 64 variables were obtained for magnitude for each patient or Control. 

From the 64 variables for latency, only two were found to be statistically different, with *p* < 0.05, shown in Table 2. These two variables were in two different bands, Delta and Alpha. The variable in the Delta band was located in OZ (*p* < 0.01), and the one in the Alpha band was situated in PO7 (*p* < 0.01). The locations of these two electrodes are shown in Figure 7a.

The power latency had the least variables. Only two electrodes in one band each were found to have statistical differences. In Figure 7b, the distribution of latencies is shown for the ALS and Control group in location OZ for the Theta band. The ALS group showed a higher latency than the Control. In Figure 7c, the Alpha band was shown in location PO7. A higher latency was seen for ALS as well.

### 3.3. Connectivity

For the connectivity analysis, all 16 electrodes were compared with all the other 15 electrodes to find their connectivity. This was performed among four bands, resulting in 960 variables for each patient. All of the 960 groups underwent the statistical analysis, and only nine variables were found to be statistically different, with *p* < 0.05. Six of these variables were in the same band, in the same electrode. 

The selected variable pairs, in this case, were located in CZ—C4 in the Delta band, PO8—OZ in the Theta band, and FZ-CP4 in the Beta band (*p* < 0.05); these three pairs are shown in Figure 8a. The other six pairs were located in the Alpha band and were all between the PO7 and another electrode (*p* < 0.05). The other pairs were C3, C1, CZ, C2, C3, and CP4, as shown in Figure 8b. This information can be seen in Table 3. 

The connectivity variables were mainly focused on the Alpha band between electrode PO7 and other electrodes in the Central area. In Figure 8c, we can see the distributions of connectivity of channels CP4 and PO7. It is clear that in the CP4 electrodes, both groups were difficult to distinguish. In contrast, in PO7, the green group (ALS) was below the Control group in almost every electrode. 

This indicates a decreased connectivity for the ALS group.

In Figure 9a–c, the following electrode pairs are seen, PO7 and CP4, PO7 and C2, and FZ and CP4, respectively. In the first two-electrode pair, a statistical difference was found in the Alpha band, which is marked in blue in Figure 9a,b. In this area, we can see a negative deflection for the ALS group. The Alpha occipital cycles are thought to be activated during temporal integration in visual perception [23]. 

In Figure 9c in the beta band, we can see that the Control group was below the ALS group in all of the bands, indicating augmented connectivity for the ALS group.

### 3.4. Classification Model

To select the variables for the classifier, first, a fourfold was performed to extract 25% of participants of each group for testing. With the remaining 75%, a threefold was performed to select the most important variables. To pick them, in each fold, it was determined which variables were significantly different between groups. This process was in a loop and was repeated 25 times. The variables selected the most times among all the repetitions were considered for the classification testing. The chosen variable was power in CZ in the Beta band. A simple linear SVM was trained with this variable, and with a leave-one-out cross-validation, a 100% in specificity and 75% in sensitivity were achieved, with an overall 89% success classifying individuals into each group. The reason of the effectiveness of this classification is shown in Figure 10a, where the ALS and control groups showed different population densities. The probability density function of the Healthy Control (HC) group and ALS group was estimated with the nonparametric kernel density estimation method. The ALS group seemed to have a bimodal distribution with one of its modes being inside the HC group. 

### 3.5. Evolution of Patients

Data were gathered of two ALS patients at different times, with three months between each session. In Figure 10b, a graph that contains the values of the *ISPC* between Cz and PO7 in the Alpha range can be appreciated. The values calculated for three sessions for patient 2 and four sessions for patient 1 can be seen. For patient 2, a reduction in connectivity was seen as the disease advanced. Patient 1 had a similar result, but a connectivity increase was seen in the third session.

## 4. Discussion

The main objective of this study was to understand the underlying cognitive neural alterations that affect people with ALS as an aid for the detection or to monitor the disease’s evolution. In the present work, alterations were found for ALS patients’ EEG data during a cognitive test, a P300 task, in comparison to HC subjects. These alterations for the ALS group were a decreased activity in the Beta band in electrode locations FZ, CZ, and PZ; around them, an augmented latency in frequency bands Delta (OZ) and Alpha (PO7); and variations in connectivity among all frequency bands, but an especially reduced band-specific connectivity in the Alpha band between channel PO7 and channels above the motor cortex (CZ, C1, C2, C3, C4, and CP4). The tracking of connectivity values in two ALS patients indicated an Alpha-related connectivity decrease between channels PO7 and CZ. Finally, data from CZ were used to classify individuals between both groups; cross-validation achieved a 100% in specificity and 75% in sensitivity, with an overall 89% success.

Decreased activity in the Beta band in electrodes over the sensorimotor band was found. Beta band activity in the sensory-motor band had been found to be important for accurate motor performance in healthy individuals [24]. This also supports what has been reported for ALS individuals [10,23,24,25]. Motor system degeneration in ALS individuals has been linked to a decrease in the Beta band [25]. This strengthens the theory that CZ could work as a biomarker to monitor ALS. The only problem is that those studies were made during a motor task, not a cognitive task. On the cognitive side, beta oscillations are also traditionally associated with sensorimotor processing [22]. This indicated a sensorimotor processing dysfunction for ALS individuals. Additionally, the Beta band is associated with attention, so it is expected to be of interest. A reduction in P300 power is usually seen in older patients, but this effect is typically present in the PZ electrode [26]. However, in the graphical analysis, the difference was detectable in CZ but not in PZ, giving a strong indication that the effect is not due to age but ALS degeneration. Moreover, when reperforming the graphical analysis only with subjects older than 50 in the control group, the region in the CZ channel was still present. 

Connectivity results indicated overactivation in the Beta band and underactivation in the Alpha band for ALS individuals. The difference in the connectivity maps could be clearly seen, and it was noticeable how these changes in connectivity were band-specific. A decrement in Alpha connectivity was so evident that an apparent valley in connectivity was located on this band between electrodes PO7 and CP4. Alpha band oscillations has been linked with a top-down control of the temporal resolution of visual perception [27]. More research is needed, and additional tests must be performed to study more deeply the connectivity in ALS patients; this could mean applying additional filtering to the signals, such as a Laplace, or calculating another type of connectivity between electrodes, such as power correlation. Additionally, the evolution of the two ALS patients whose results were available at different time points indicates an overall decrement in the connectivity between PO7 and CZ in the Alpha band, strongly indicating a connectivity degeneration in the Alpha band for ALS individuals. Alterations in connectivity in the Alpha band have been found in ALS patients [6,28] and also in the Beta band [5]; yet, these results were found in the rest-state. Beta band connectivity has also been found to be essential for accurate motor performance in healthy individuals [24]. This may indicate an over-effort from ALS individuals, but as the connectivity results in the Beta band were only significant in one pair of electrodes, this is hard to generalize. A reduction in connectivity has been linked to a decrease in cognition for patients with Multiple Sclerosis [29]. Rates of ALS-related impairment are noted to be related to the disease stage. Cognitive deficit is more frequent with more severe ALS stages [30]. This strengthens the theory that connectivity could serve as a tool to monitor the disease’s advances in cognitive atrophy, specifically. 

Most of the previous papers that have studied either connectivity or signal amplitudes did so in the resting-state, not during an active task [6,7,23,28], or during a motor task [10,24,25]. The few that have studied it during a P300 task did not analyze connectivity and amplitude simultaneously in a spectral analysis [11]. The selected variable for classification was the peak power in the Beta band in CZ. Achieving a 100% in sensitivity indicates that no false positives were present, which strengthens the possibility of using this characteristic as a potential biomarker to track ALS degeneration. Classifiers for ALS individuals have been performed, but they do not usually analyze the signals [31]. Using cognitive alterations in the Beta band to classify between ALS and Control groups has been performed with success but using a generic BCI (BCI2000) [20]. The signals used in this work came from a P300-based BCI for ALS patients [21], so understanding the cognitive neural alterations occurring could help improve the BCI’s performance as its performance will most likely be different for a CLIS ALS patient than for a control subject, or even a more moderate ALS case. A follow-up of an ALS patient’s potential biomarker has not been performed to our knowledge. For the magenta patient in Figure 10b, a gradual decrement was seen for the peak power in the Beta band in CZ. Yet, this was not the case for the other ALS patients represented. This may be because the magenta patient had a more severe case of atrophy, but without further research, this is mere speculation. The results were achieved with a simple univariate Support Vector Machine (SVM) classification model. The only three patients that were classified as Controls had either not very good signals or had taken the P300 evaluation previously, which may alter the results. 

In conclusion, the ALS group seemed to have a statistically important difference in power and connectivity during the P300 task, a cognitive task, the two most important being in power magnitude in the Beta band and connectivity in the Alpha band. All 21 variables had *p* > 0.05 after FDR correction. Yet, the evidence implied that some of these results may be false negatives. This evidence is the fact that the Cz, C2, and C4 electrodes showed significant differences in the same region (Beta band around 500 ms after stimuli) in the time–frequency power maps, and that the power values of electrode Cz in the Beta band had a good overall performance in classifying correctly between ALS and HC groups. This is also the case for the connectivity decrease for the ALS group in the Alpha band, as Figure 9 clearly shows a frequency band-related decrease, and the connectivity value decreased in ALS patients (in the same electrode pair) as the disease advanced. The variables selected by the analysis did not seem to be random but had a correlation with what other researchers have found, strengthening the theory that they could serve as biomarkers for ALS. The SVM model that resulted from the classification between ALS patients and Control subjects had very promising results. All Control subjects were classified correctly, which means that a false ALS diagnosis would not occur. Clearly, this was a simple bivariate model. A much more complex model may be obtainable. The potential of these variables as ALS biomarkers that could aid detection or monitor the advance of the disease is noticeable and must be further studied. Finally, this was an exploratory research whose objective was to find areas that potentially need further examination. The hypothesis for the underactivations presented by the ALS group is a general neural degeneration; more studies are needed to localize the degeneration site and the level of its affectations. The next stage is to test these results by observing these specific variables with more electrodes in these areas.

## Figures and Tables

**Figure 1 sensors-21-06801-f001:**
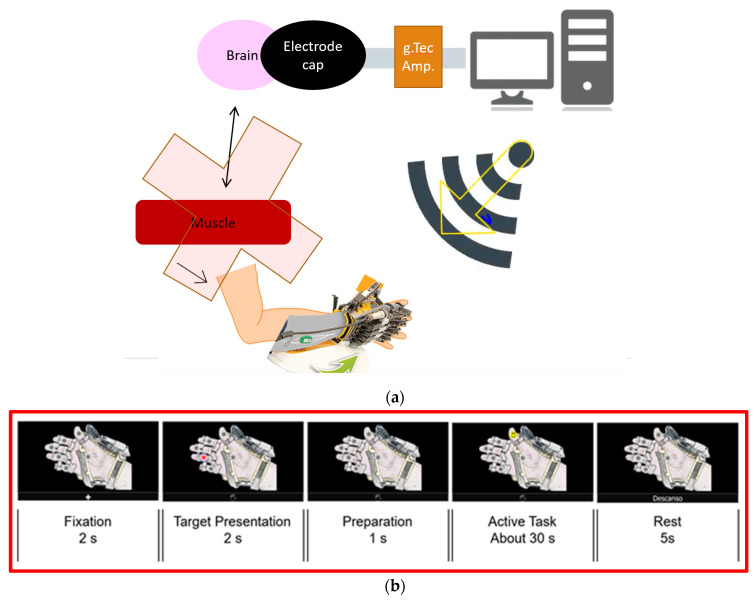
(**a**) Layout of the BCI (Brain Computer Interface) system whose purpose is to generate an alternative pathway for muscle movement. (**b**) Graphical interface of the P300-BCI in its different stages used to control a robotic hand-orthosis [21]. (**c**) Position of electrodes used in this P300-BCI according to 10–20 system.

**Figure 2 sensors-21-06801-f002:**
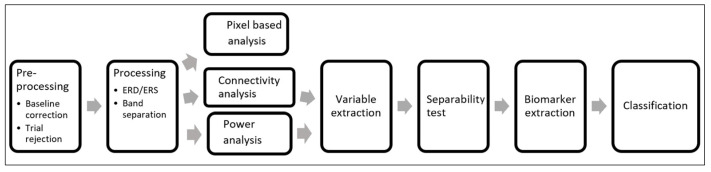
Processing stages of information.

**Figure 3 sensors-21-06801-f003:**
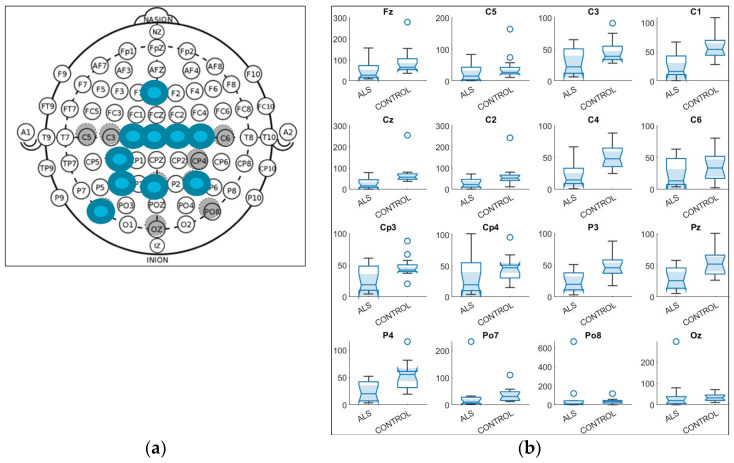
(**a**) Electrodes’ locations found to be statistically different in power magnitude variables between Amyotrophic Lateral Sclerosis (ALS) ALS and Control group; (**b**) distribution of power magnitude variables for each electrode of ALS group and Control group.

**Figure 4 sensors-21-06801-f004:**
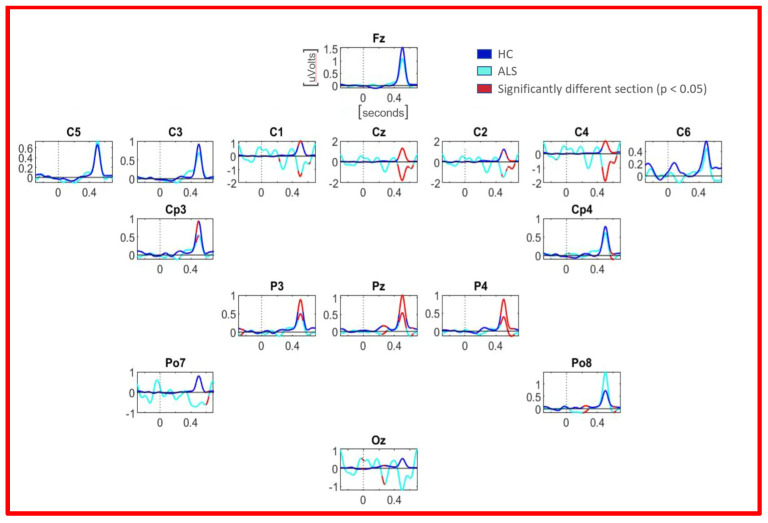
Mean ERPs of all 16 channels in Beta band. In cyan, the ALS group mean is observed, while blue represents the Control group. The segments that are red are significantly different with *p* < 0.05. The *x*-axis represents milliseconds and *y*-axis microvolts.

**Figure 5 sensors-21-06801-f005:**
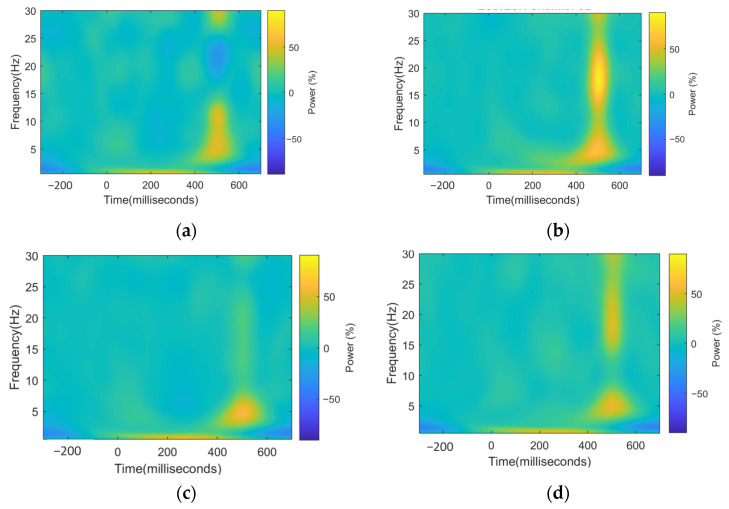
(**a**) Mean spectral power percent change of electrode CZ for ALS group; (**b**) mean spectral power percent change of electrode CZ for Control group; the Control group has a power increment and the ALS group has a power decrement; (**c**) mean spectral power percent change in electrode PZ for ALS group; (**d**) mean spectral power percent change in electrode PZ for Control group.

**Figure 6 sensors-21-06801-f006:**
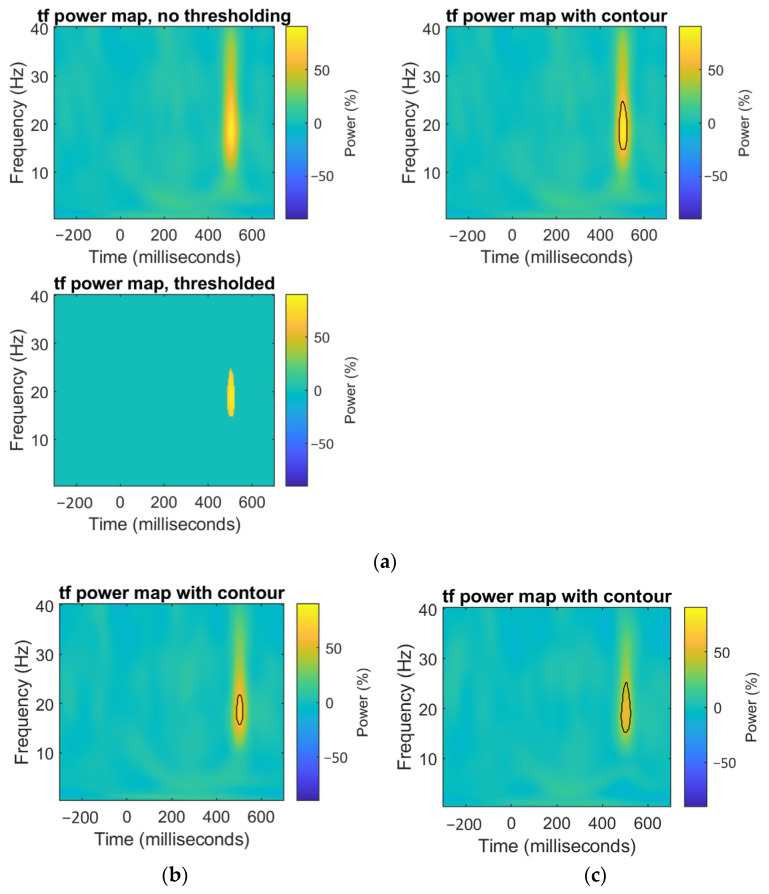
(**a**) Channel CZ. Time–frequency power map (up left). Time–frequency power map with the significant area highlighted (up right). Time–frequency power map displaying only significantly different area (down left); (**b**) Channel C2. Time–frequency power map with significant area highlighted; (**c**) Channel C4. Time–frequency power map with the significant area highlighted.

**Figure 7 sensors-21-06801-f007:**
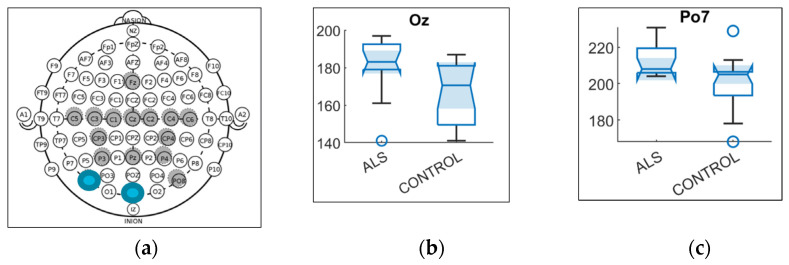
(**a**) Electrodes’ locations found to be statistically different in power latency variables between ALS and Control group; (**b**) distribution of latencies for ALS and Control group in location OZ for the Theta band; (**c**) distribution of latencies for ALS and Control group in location PO7 for the Alpha band.

**Figure 8 sensors-21-06801-f008:**
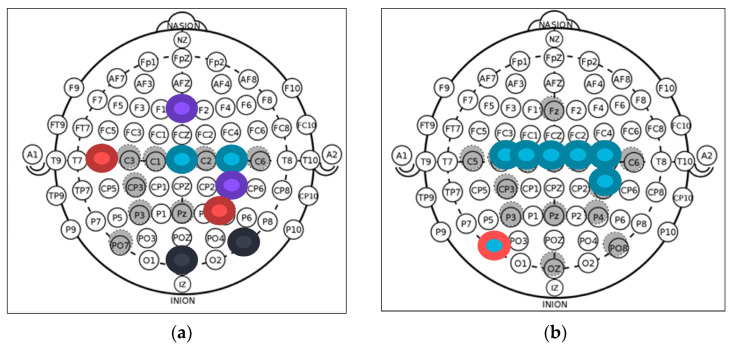
(**a**) Electrodes’ locations found to be statistically different in connectivity variables between ALS and Control group for bands Delta, Theta, and Beta; (**b**) electrodes’ locations found to be statistically different in connectivity variables between ALS and Control group for Beta band; (**c**) *ISPC* of all subjects and patients of electrodes PO7 (bottom) and CP4 (top) with all channels. Control is in blue and ALS in green. Most connectivity values’ significant differences were found between the PO7 electrode and the electrodes above the motor cortex area, CZ, C1 C2, C3, C4, and CP4. This can be seen in the PO7 chart, where almost all ALS connectivities are below the Control group.

**Figure 9 sensors-21-06801-f009:**
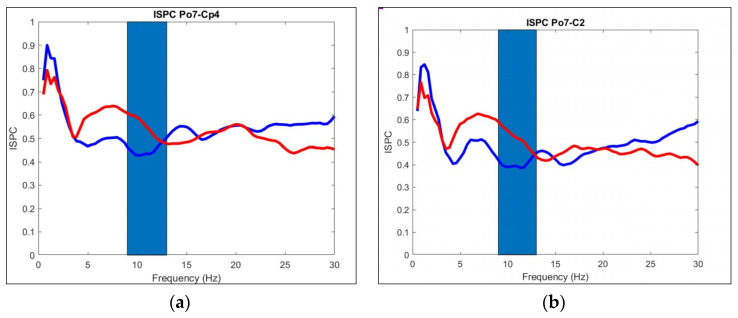
(**a**) Average InterSite Phase Clustering (*ISPC*) between electrodes PO7 and CP4 of Control (red) and ALS (blue). Alpha band is shown in blue. In the Alpha band, the ALS group is below the Control group; (**b**) Average *ISPC* between electrodes PO7 and C2 of Control (red) and ALS (blue). Alpha band is shown in blue. In the Alpha band, the ALS group is below the Control group; (**c**) Average *ISPC* between electrodes FZ and CP4 of Control (red) and ALS (blue). Beta band is shown in blue. In the Beta band, the ALS group is above the Control group.

**Figure 10 sensors-21-06801-f010:**
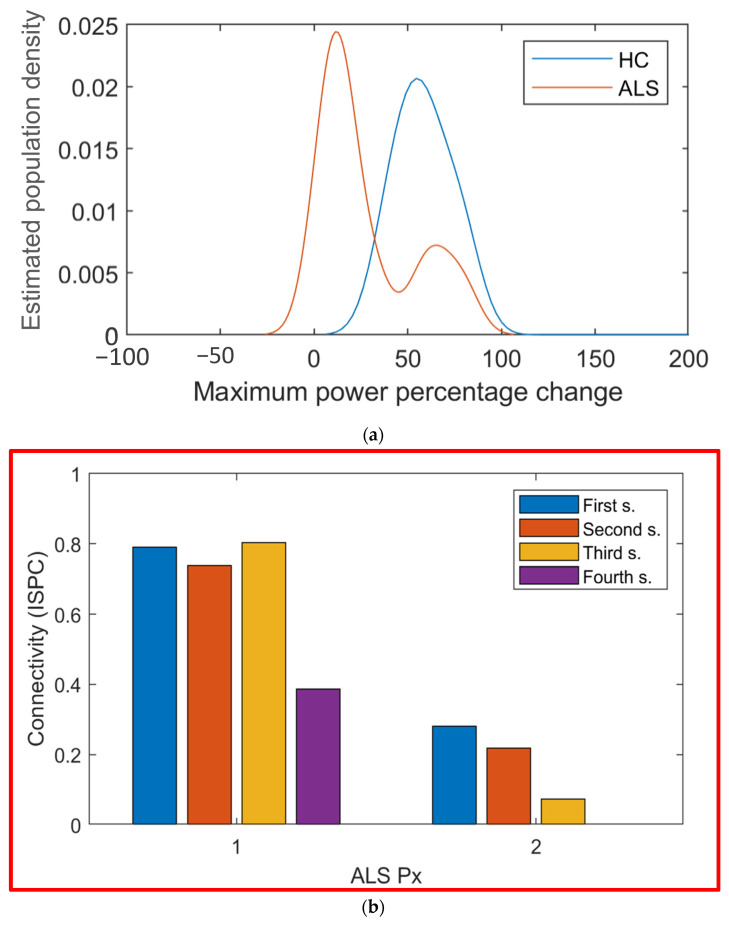
(**a**) Estimated population density of maximum power for ALS group and Healthy Control; (**b**) connectivity (*ISPC*) value between electrodes CZ and PO7 in the Alpha range for four different sessions for Px1 and three different sessions for Px2. Each session was three months apart.

**Table 1 sensors-21-06801-t001:** Variables of power magnitude with significant difference resulting from a ranksum test (Wilcoxon for independent groups) between ALS group and Control group.

ELECTRODE	*p*-Value
BETA	
FZ	0.0244
C1	0.0013
CZ	0.0028
C2	0.0114
C4	0.0028
CP3	0.0434
P3	0.005
PZ	0.0066
P4	0.0043
PO7	0.0148

**Table 2 sensors-21-06801-t002:** Variables of power latency with significant difference resulting from a ranksum test (Wilcoxon for independent groups) between ALS group and Control group.

ELECTRODE	*p*-Value
DELTA	
OZ	0.015
ALPHA	
PO7	0.011

**Table 3 sensors-21-06801-t003:** Variables of connectivity with significant difference resulting from a ranksum test (Wilcoxon for independent groups) between ALS group and Control group.

ELECTRODE PAIR	*p*-Value
DELTA	
CZ—C4	0.0484
THETA	
PO8—OZ	0.0274
ALPHA	
PO7—C3	0.0484
PO7—C1	0.0274
PO7—CZ	0.0484
PO7—C2	0.0346
PO7—C4	0.0484
PO7—CP4	0.0308
BETA	
FZ—CP4	0.0434

## Data Availability

The datasets generated and analyzed for this study are available upon request to the corresponding author.

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
