# Peer review of "Functional Connectivity and Frequency Power Alterations during P300 Task as a Result of Amyotrophic Lateral Sclerosis"

_sensors, 2021, doi:10.3390/s21206801_

Round 1

Reviewer 1 Report

I was happy to review this study that aimed at determining neurophysiological markers of Amyotrophic Lateral Sclerosis (ALS) using EEG. A group of individuals with ALS and a control group were submitted to an Oddball task while their brain activity was recorded. The authors then applied cross-validation methods to discriminate between ALS and control participants based on the neurophysiological measures that proved that theses measures can be markers of ALS.

The article is well written, although clarity and grammar can be improved throughout the manuscript. The use of the Future tense seems inappropriate. Avoid sentences without a verb. Do not use abbreviations (e.g., L551)

I am a little confused about the FDR correction. I understand that tests are considered significant when the p-value is below .056. That value is over the usual Alpha level (.05) used in the literature for one comparison. In point 3.1.3 FDR correction, it is mentioned that “Only the power results re-364 main slightly significant after the FDR correction with a p-value of 0.056”. Is that true for results presented above in the manuscript? In table 3, significant values are very close to .05, which means that they might be obtained by chance given the number of comparisons. Please clarify these points.

I am not an expert in the EEG methods presented here, but it seems to me that only 16 electrodes is not enough to perform connectivity analyses. Figure 1e presents a much bigger EEG set; I understand that only electrodes presented in grey were used. Can the author clarify why they used only those electrodes?

Introduction

The oddball task should be briefly described in the introduction. It could be through an example of what the participant sees and the task they have to perform.

L72: the sentence is not grammatically correct

Alpha and Beta bands should also be briefly described

I do not think the last paragraph of the introduction is necessary. The article follows a typical scientific writing organization

Materials and Methods

P300-BCI System

I really have difficulties understanding the task. It is not clear why the participant wears a robotic hand-orthosis. In the description of the task, it seems that there is no movement required, the participant has to simply pay attention to the stimuli. What are the happy the faces, why are they appearing? An additional picture of the screen that the participant sees, and how that connects to the orthosis would help.

Data acquisition

I do not think the first sentence belongs in this section.

Subjects

What do the authors mean by “the significance of selected variables.” (L152)?

Pre-processing

L174: Use a comma instead of a period, otherwise the next sentence has no verb

L244: The sentence has no verb

Separability test

Last sentence: it is awkward to start a sentence with “But” in this context. Please Reformulate

FDR correction

The last sentence is not clear.

Extreme pixel correction

L293: does non-statistical mean not significant?

Results

No mention of Figure 3(a) is made in the text, I understand that” Figure 4(a)” (L309) should be Figure 3(a)

Distribution of power variables

L336-337: the sentence has no verb

Discussion

L507-508: the sentence has no verb

L521-522: “More research is needed, and additional tests must be performed to study the real connectivity in ALS patients.” What do the authors mean by “real connectivity”? Is not the connectivity tested in this study real? What additional tests should be performed?

L537-539. The sentence is not grammatically correct. It is awkward to include a paragraph with only one sentence. It is not clear what are topic of this paragraph and the previous one. Please modify the logic.

L559: “importance” should be “important”

Reviewer 2 Report

This work investigates the oscillatory dynamics of scalp EEG responses from patients with amyotrophic lateral sclerosis (ALS) and healthy controls (HC). Patients and participants both completed a brain-computer interface (BCI)-robotic arm experiment where they imagined and counted finger movements under an oddball condition. The proposed methodology is sophisticated and therefore could be used in future BCI research. I also found the findings convincing. However, some details need clarification. I suggest a revision at this point. Concerns/issues along with some suggestions are listed below,

General/major issues,

  • The use of the term, synchrony or desynchrony, is a bit controversial (not agreed by all) these days. In brief, it is not clear whether the relative power changes truly represent (de) synchronization of neural ensembles. Besides, it can be confused with the term connectivity, which investigates interactions (synchronization) between two (or more) brain areas. I suggest using the term power frequency decrease or increase (changes) throughout the manuscript.
  • I also suggest avoiding implications regarding cortical regions. Due to the known issue of volume conduction (one scalp area is the result of multiple source areas), scalp-level activations cannot be ascribed to a particular cortical area.
  • Some details in the experiment section are missing. Specifically, it is not clear whether the task involved actual motor activities, or it was all imagery, also not clear how counting was conducted (loud or quiet/receptive). And, more importantly, what was the motivation for using the palm happy face trials only and discarding other cues/responses?
  • Some implications of the findings are missing or have not been clearly explained. There is no clear hypothesis about the underlying neural dynamics of the task activation (what to expect from that data condition, happy face?). Although the phenomenon of frequency power changes is not fully known, power decrease effects, especially in the range of beta, have been linked to sensory-motor integration processes, whereas increased beta (aka. rebound), is more connected to post-movement motor activities – see works by Pfurtscheller. Also, from the time-frequency analyses (Figs. 5-6), there is a strong power increase in the range of beta around 500-ms. Was that due to (imagination or actual) hand-movement or was it due to pre-or post-movement activations? Implications for such effects are missing.
  • Some used/created terminologies seem inconsistent/confusing, e.g., pixel-wise/based, or variable/biomarker extraction. I suggest using better/simpler terms such as time-frequency analysis, feature extraction, or cluster-based statics.
  • Most figures lack proper labeling, units, and legends (see below for more details).

Minor,

  • Abstract
    • Results have been reported only for the ALS group; results for healthy control (HC) are missing. I also think the findings from HC should be used as a benchmark for time-frequency analysis. What could make these analyses more interesting was to investigate the differences in time-frequency responses between HC and ALS groups.
    • Some details are confusing and thus require revision. e.g., "increased latency in Delta (OZ)"; does the author mean latency in ERP responses in the range of the delta band? It seems confusing.
  • Results/Figures,
    • 4. Please add units and legends to the figure. Findings should be presented for both ALS and HC.
    • Figures 5-6. Please add units and a colorbar. The Beta power decrease reported in the Abstract section is not shown here.
    • In figure 10, the slope of the separation function is almost flat. This means there is almost no statistical relationship between alpha and beta power values. I am surprised to see such high classification accuracy.

I hope this helps.

Round 2

Reviewer 2 Report

The authors have successfully addressed my comments and concerns. I believe the work is sufficient for publication.